# Synthesis and Evaluation of Trypanocidal Activity of Chromane-Type Compounds and Acetophenones

**DOI:** 10.3390/molecules26237067

**Published:** 2021-11-23

**Authors:** Luis A. González, Sara Robledo, Yulieth Upegui, Gustavo Escobar, Wiston Quiñones

**Affiliations:** 1Grupo de Química Orgánica de Productos Naturales, Instituto de Química, Facultad de Ciencias Exactas y Naturales, Universidad de Antioquia, Calle 70 N° 52-21, Medellín 050014, Colombia; luis.gonzalez12@udea.edu.co (L.A.G.); gustavo.escobar1@udea.edu.co (G.E.); 2Programa de Estudio y Control de Enfermedades Tropicales (PECET), Facultad de Medicina, Universidad de Antioquia, Calle 70 N° 52-21, Medellín 050014, Colombia; sara.robledo@udea.edu.co (S.R.); yulieth.upegui@udea.edu.co (Y.U.)

**Keywords:** *Trypanosoma cruzi*, chromane, phenolic compounds, prenyl derivatives

## Abstract

American trypanosomiasis (Chagas disease) caused by the *Trypanosoma cruzi* parasite, is a severe health problem in different regions of Latin America and is currently reported to be spreading to Europe, North America, Japan, and Australia, due to the migration of populations from South and Central America. At present, there is no vaccine available and chemotherapeutic options are reduced to nifurtimox and benznidazole. Therefore, the discovery of new molecules is urgently needed to initiate the drug development process. Some acetophenones and chalcones, as well as chromane-type substances, such as chromones and flavones, are natural products that have been studied as trypanocides, but the relationships between structure and activity are not yet fully understood. In this work, 26 compounds were synthesized to determine the effect of hydroxyl and isoprenyl substituents on trypanocide activity. One of the compounds showed interesting activity against a resistant strain of *T. cruzi*, with a half effective concentration of 18.3 µM ± 1.1 and an index of selectivity > 10.9.

## 1. Introduction

Parasitic infections are the main causes of morbidity and mortality in the world; in addition, the emergence and re-emergence of many of these parasite diseases are accelerated by climate change, the increasing migration and the increasing parasite resistance to drugs, among others, contribute to the deterioration of global public health [1].

Chagas disease, also known as American trypanosomiasis, is the result of infection by the *Trypanosoma cruzi* (*T. cruzi*) parasite [2]. Transmission of this parasite to humans occurs primarily through triatomine bugs, although it can occur through the placenta, blood transfusion, organ transplantation, or consumption of food contaminated with triatomine feces. It is estimated that in the world there are between 6 to 7 million people infected with *T. cruzi* but the regions with the highest prevalence are the rural areas of South America and Central America [2].

Today, there are constraints to using the available drugs against Chagas disease, nifurtimox, and benznidazoles (BNZ), due mainly to bioavailability and efficacy during the chronic phase of the disease. Besides, low treatment adherence due to severe side effects such as vomiting, anorexia, peripheral neuropathy, and allergic dermopathy has been noticed [3].

Some natural products such as acetophenones and chalcones, as well as chromane-type substances, such as chromones and flavones, exhibit trypanocide activity [4,5,6,7,8]. In these compounds, besides the phenolic hydroxyls are also frequently present one or several free or cyclized isoprenyl units. Nonetheless, very little is known about the contribution of these groups to the biological activity of the compounds. In this work were synthesized 26 molecules to study the importance of these functional groups in the trypanocidal activity. The effect of the hydroxyl and isoprenyl substituents and pyran-4-one systems in the trypanocidal activity was also determined.

## 2. Results

### 2.1. Compound Synthesis

The compounds **5**–**30** were prepared by adapting the procedures previously reported elsewhere [9,10,11,12,13,14,15]. The reaction sequence is shown in Figure 1. Purification of the products was carried out through liquid−liquid extraction and chromatographic separations. Yields were 6.3–77.1%; due to low yield processes, reactions **II**–**VII** were repeated at least three times. Compounds were identified by ^1^H-NMR, ^13^C-NMR, and HRMS (see Appendix A online for details). Although compounds **5**–**30** have not been tested against *T. cruzi*, the search for chemical compounds using a structure search in SciFinder showed that only **19**–**30** do not have any preliminary reports.

### 2.2. In Vitro Cytotoxicity and Trypanocidal Activity

All compounds except **23**–**25** showed high cytotoxic activity to human U937 macrophages. BNZ showed moderate activity against *T. cruzi* and no cytotoxicity with values of EC_50_ of 56.5 ± 1.5 µM and an LC_50_ > 768.5 µM, respectively (Table 1). Compounds **6**, **16**, **18**, and **28** showed high cytotoxicity, with LC_50_ < 10 µM and especially **28**, which showed similar activity to DOX, with LC_50_ of 0.5 ± 0.12 µM.

Twenty-four of the twenty-six compounds evaluated were more effective than BNZ with EC_50_ values ranging from 2.6 µM (**18**) to 49.3 µM (**9**), with compounds **6**, **7**, **17**, **18**, and **26** exhibiting an EC_50_ < 10 µM, of which **18** and **6** were the most active with EC_50_ values of 2.6 µM and 3.3 µM, respectively. Compounds **8**, **9**, **11**, **12**, **15**, **25**, and **27** showed moderate activity, with EC_50_ values varying between 11.8 µM and 49.3 µM. Only compounds **13** and **24** showed very low activity, with EC_50_ values of 154.4 µM and 111.1 µM. For compounds **10**, **14**, **16**, **23**, and **28–30**, the effective concentration for *T. cruzi* exceeded the concentration that was toxic to the host cells, and therefore the exact EC_50_ value could not be determined and therefore is reported with the sign “greater than”.

When trypanocidal activity and cytotoxicity were correlated in compounds with better EC_50_, such as **6**, **16**, **18**, and **28**, very unfavorable IS were found, usually below 2. In contrast, compound **8** lacking isoprenyl was the most promising, with an IS > 10.9 (Table 1).

## 3. Discussion

This work describes the synthesis of derivatives of acetophenones, chromones, chalcones, and flavones, with hydroxyl and isoprenyl substituents. These types of molecules are of natural origin and present a wide spectrum of reported biological activities, but the knowledge on how the structural features of these molecules are involved in the interaction with a possible receptor is reduced. Therefore, 26 synthetic derivatives (six series) of these substances were evaluated in vitro against *T. cruzi*, analyzing mainly the effect on the biological activity of heterocycles condensed to the aromatic rings (pyran-4-one system) and the hydroxyl and isoprenyl substituent groups. The series evaluated were: series A, polyhydroxy-substituted acetophenones **5**–**9**; series B, hydroxy-substituted chromones **10**–**13**; series C, hydroxy-substituted and isoprenylated acetophenones **14**–**18**; series D, hydroxy-substituted and isoprenylated chromones **23**–**26**; series E, isoprenylated chalcones **27** and **28;** and, series F, isoprenylated flavones **29** and **30**.

All compounds of series C (hydroxy-substituted and isoprenylated acetophenones) showed good trypanocidal activity, especially **16**–**18** when compared to compounds **5**–**9** (series A) that corresponded to polyhydroxy-substituted acetophenones. Apparently, the introduction of the isoprenyl group plays an important role in modulating the polarity of the molecule which is corroborated by noting the differences in activity between **18** and **8** (2.6 ± 0.1 μM vs. 18.3 ± 1.1 μM). However, because in the case of isoprenylated chromones **23**–**26** where no gain in activity with prenylation was observed, other factors than polarity must be involved in the activity.

On the other hand, the presence of a hydroxy in the position adjacent to the carbonyl (**16** and **17**) also increases the activity as seen in compound **7**, one of the most active compounds of series A (LC_50_ = 10.5 ± 0.6 μM). This effect is possibly caused by increased nucleophilicity towards the carbonyl group due to the inducing effects of the carbonyl or its stabilization by an additional hydrogen bridge.

In the chromones (**10**–**13** series B and **23**–**26** series D), chalcones and flavonoid compounds (**27** and **28** series E and **29** and **30** series F), the activity is only marginal, except for compound **28** which has in addition to a p-chlorophenyl group and the αβ-unsaturated carbonyl group which has already been demonstrated to be important in antiparasitic activity [16].

In general, the toxicity of the active compounds tested here against *T. cruzi* was high. Despite this, the correlation between trypanocidal activity and cytotoxicity was good, with IS greater than 1 and even greater than 10.9. However, although cytotoxicity is usually determined as an important parameter to evaluate in the search for substances with therapeutic potential, it should be noted that in vitro cytotoxicity tests only reflect the effect on the specific cell type used in the test, and not on an entire organism, which, as in animals, has a digestive system and metabolism mechanisms that can modify the toxicity of a substance. Hence, both the trypanocidal activity and toxicity of a substance must be confirmed in in vivo studies using the respective disease models. Furthermore, these molecules still need to be optimized in their structure, in order to try to increase their activity and modulate bioavailability.

In summary, compounds **6**, **7** and **8** (substituted dihydroxy acetophenones), **11**, **12**, **25** and **26** (chromones), **16**–**18** (isoprenylated acetophenones), chalcone **28** and flavanones **29** and **30** showed much better trypanocidal activity than the control drug BNZ. The trypanocidal activity identified in these compounds, even better than that reported for BNZ, selects them as “hit” compounds to start the development of drug candidates for the treatment of *T. cruzi* infection. However, we must proceed not only with the validation of this activity in vivo models but also with target-identification and mechanism-of-action studies to confirm specific bioactivity, given that some of these chemical structures can be related to pan-assay interference compounds (PAINS) [17].

## 4. Materials and Methods

### 4.1. Chemistry

All commercially available reagents and solvents were obtained from commercial suppliers and used without further purification. Phenol derivatives (**1**–**4**) were purchased from Sigma Chemical Co. (St. Louis, MO, USA). Thin-layer chromatography (TLC) with silica gel 60 F254-impregnated aluminum sheets (0.25 mm, Merck, Darmstadt, Germany) was used to check the progress or reactions, and compounds were detected under spraying with vanillin (3% in H_2_SO_4_) and heating at 110 °C. The chromatographic separations were performed using preparative column chromatography with silica gel 60 (200–300 mesh, Merck, Darmstadt, Germany). The melting points were determined using a Mel-Temp apparatus (Electrothermal, Staffordshire, UK). The ^1^H, ^13^C, and 2D NMR spectra of the synthetic compounds were recorded on a Bruker Fourier 300 spectrometer (Bruker Bio-Spin GmbH, Rheinstetten, Germany) operating at 300 MHz for ^1^H and 75 MHz for ^13^C NMR, using CDCl_3_ (Sigma, St Louis, Mo, USA) as the solvent, and TMS as an internal standard. Chemical shifts (δ) are reported in ppm, and the coupling constants (*J*) are reported in Hz. High-resolution mass spectra were obtained using an ultra-high resolution Qq-time-of-flight (UHR-QqTOF) mass spectrometer (Impact II-Bruker), with an electrospray ionization source in positive ion mode. 

### 4.2. General Procedure for Preparation of 2-Hydroxyacetophenone Derivatives (I)

The synthesis of **5**–**9** was conducted according to literature [9]. Thus, phenol derivatives (**1**–**4**) (13.2 mmol) dissolved in acetic anhydride (Ac_2_O) (3 mL, 30 mmol) were added in ethyl acetate (AcOEt) (5 mL). Then, boron trifluoride-diethyl ether (BF_3_-Et_2_O, 800 µL, 6.4 mmol) was slowly added to the reaction mixtures, a reflux setup allowed the mixtures to be heated in a controlled manner at 50 °C for 12 h without the loss of solvent. The work-up process was performed using 100 mL of water, neutralization with NaHCO_3_, and extraction with CH_2_Cl_2_ (3 × 100 mL). The organic phase was brought to dryness and the resulting solid was eluted on a silica gel 60 columns using hexane:ethyl acetate (Hex:AcOEt) (2:1) (*v:v*). The synthesis of **5**–**9** was achieved in an overall yield of 16.5–56%.

#### 4.2.1. Compound (**5**)

1-(2,4-Dihydroxyphenyl)ethan-1-one (**5**). Yield 30.4%, yellow solid, m.p.: 144–145 °C. ^1^H NMR (300 MHz, CDCl_3_) δ 7.56 (d, *J* = 8.8 Hz, 1H), 6.35 (dd, *J* = 8.8, 2.4 Hz, 1H), 6.29 (d, *J* = 2.3 Hz, 1H), 2.50 (s, 3H). ^13^C NMR (75 MHz, CDCl_3_) δ 202.71, 164.72, 164.64, 133.03, 113.38, 108.34, 102.91, 26.10. HRMS (ESI) *m*/*z*, calculated for C_8_H_9_O_3_ [M+H]^+^ 153.0546; found 153.0547.

#### 4.2.2. Compound (**6**)

1-(2,5-Dihydroxyphenyl)ethan-1-one (**6**). Yield 26.3%, yellow solid, m.p.: 200–204 °C. ^1^H NMR (300 MHz, CDCl_3_) δ 7.13 (d, *J* = 2.9 Hz, 1H), 7.00 (dd, *J* = 8.9, 2.9 Hz, 1H), 6.78 (d, *J* = 8.9 Hz, 1H), 2.54 (s, 3H). ^13^C NMR (75 MHz, CDCl_3_) δ 204.48, 155.41, 148.76, 125.04, 119.35, 118.76, 115.34, 26.67. HRMS (ESI) *m*/*z*, calculated for C_8_H_9_O_3_ [M+H]^+^ 153.0546; found 153.0546. 

#### 4.2.3. Compound (**7**)

1-(2,6-Dihydroxyphenyl)ethan-1-one (**7**). Yield 49.1%, yellow solid, m.p.: 160–162 °C. ^1^H NMR (300 MHz, CDCl_3_) δ 7.20 (t, *J* = 8.2 Hz, 1H), 6.40 (d, *J* = 8.2 Hz, 2H), 2.74 (s, 3H). ^13^C NMR (75 MHz, CDCl_3_) δ 205.83, 161.97, 136.30, 110.47, 108.04, 33.64. HRMS (ESI) *m/z*, calculated for C_8_H_9_O_3_ [M+H]^+^ 153.0546; found 153.0546.

#### 4.2.4. Compound (**8**)

1-(2,3,4-Trihydroxyphenyl)ethan-1-one (**8**). Yield 56.0%, yellow solid, m.p.: 170–172 °C. ^1^H NMR (300 MHz, CDCl_3_) δ 7.14 (d, *J* = 8.9 Hz, 1H), 6.36 (d, *J* = 8.9 Hz, 1H), 2.46 (s, 3H). ^13^C NMR (75 MHz, CDCl_3_) δ 203.50, 151.85, 151.43, 131.83, 123.10, 113.62, 107.47, 26.07. HRMS (ESI) *m/z*, calculated for C_8_H_9_O_4_ [M+H]^+^ 169.0495; found 169.0494.

#### 4.2.5. Compound (**9**)

1-(2,4,6-Trihydroxyphenyl)ethan-1-one (**9**). Yield 16.5%, yellow solid, m.p.: 137–138 °C. ^1^H NMR (300 MHz, Acetone) δ 5.92 ppm (s, 2H), 2.60 (s, 3H). ^13^C NMR (75 MHz, Acetone) δ 203.10, 164.91, 164.90, 104.79, 95.07, 32.17. HRMS (ESI) *m/z*, calculated for C_8_H_9_O_4_ [M+H]^+^ 169.0495 found 169.0400.

### 4.3. General Procedure for Prenylation of 2-Hydroxyacetophenone Derivatives (III)

Into round-bottom flasks (10 mL) the 2-Hydroxyacetophenone derivatives **5**–**9** (4 mmol) were dissolved individually in 3-Methyl-2-buten-1-ol (2 mL, 20 mmol). The resulting mixtures were stirred at 50 °C for 15 min to obtain a full homogenization. Then, the reaction mixtures were poured into ice-water, and BF_3_-Et_2_O (150 µL, 1.20 mmol) was added dropwise [10]. After the addition of BF_3_.Et_2_O, the mixtures were then stirred at room temperature for 24 h. The work-up process consisted of an addition of 50 mL water, and successive extractions with dichloromethane. The organic layer was dried over sodium sulfate and concentrated to dryness. The compounds **14**–**18** were purified using column chromatography with silica gel 60, eluting with Hex:AcOEt (2:1) (*v*/*v*). The synthesis of **14**–**18** has been achieved in an overall yield of 5.1–12.3%.

#### 4.3.1. Compound (**14**)

1-(2,4-Dihydroxy-5-(3-methylbut-2-en-1-yl)phenyl)ethan-1-one (**14**). Yield 7.6%, white solid. m.p.: 137 °C. ^1^H NMR (300 MHz, Acetone) δ 12.63 (s, 1H), 9.52 (s, 1H), 7.60 (s, 1H), 6.36 (s, 1H), 5.41–5.26 (m, 1H), 3.26 (d, *J* = 7.2 Hz, 2H), 2.53 (s, 3H), 1.72 (s, 6H). ^13^C NMR (75 MHz, Acetone) δ 202.67, 163.46, 162.49, 132.20, 131.90, 122.57, 120.29, 113.12, 102.24, 27.53, 25.35, 25.00, 16.96. HRMS (ESI) *m/z*, calculated for C_13_H_17_O_3_ [M+H]^+^ 221.1172 found 221.1173.

#### 4.3.2. Compound (**15**)

1-(2,4-Dihydroxy-3-(3-methylbut-2-en-1-yl)phenyl)ethan-1-one (**15**). Yield 6.3%, white solid. m.p.: 148 °C. ^1^H NMR (300 MHz, Acetone) δ 13.11 (s, 1H), 9.35 (s, 1H), 7.63 (d, *J* = 8.7 Hz, 2H), 6.50 (d, *J* = 8.7 Hz, 2H), 5.33–5.16 (m, 2H), 3.35 (d, *J* = 7.2 Hz, 2H), 1.77 (s, 3H), 1.64 (s, 3H), 1.29 (s, 3H). ^13^C NMR (75 MHz, Acetone) δ 203.02, 162.78, 161.79, 130.64, 130.38, 122.32, 114.97, 113.24, 107.12, 25.36, 25.01, 21.29, 17.03. HRMS (ESI) *m/z*, calculated for C_13_H_17_O_3_ [M+H]^+^ 221.1172 found 221.1177.

#### 4.3.3. Compound (**16**)

1-(2,6-Dihydroxy-3-(3-methylbut-2-en-1-yl)phenyl)ethan-1-one (**16**). Yield 9.1%, white solid. m.p.: 75–76 °C. ^1^H NMR (300 MHz, CDCl_3_) δ 10.75 (s, 1H), 9.25 (s, 1H), 7.12 (d, *J* = 8.3 Hz, 1H), 6.33 (d, *J* = 8.3 Hz, 1H), 5.34–5.19 (m, 1H), 3.27 (d, *J* = 7.2 Hz, 2H), 2.74 (s, 3H), 1.76 (s, 3H), 1.74 (s, 3H). ^13^C NMR (75 MHz, CDCl_3_) δ 206.01, 160.21, 159.17, 136.46, 134.73, 121.82, 119.72, 110.29, 107.12, 33.68, 28.50, 25.93, 17.94. HRMS (ESI) *m/z*, calculated for C_13_H_17_O_3_ [M+H]^+^ 221.1172, found 221.1170. 

#### 4.3.4. Compound (**17**)

1-(2,4,6-Trihydroxy-3-(3-methylbut-2-en-1-yl)phenyl)ethan-1-one (**17**). Yield 10.1%. white solid. m.p.: 174 °C. ^1^H NMR (300 MHz, CDCl_3_) δ 5.84 (s, 1H), 5.21 (t, *J* = 6.9 Hz, 1H), 3.26 (d, *J* = 6.9 Hz, 2H), 2.64 (s, 3H), 1.77 (s, 3H), 1.68 (s, 3H). ^13^C NMR (75 MHz, CDCl_3_) δ 203.93, 164.31, 162.13, 160.53, 132.45, 122.66, 106.58, 104.81, 94.89, 32.59, 25.70, 21.27, 17.70. HRMS (ESI) *m/z*, calculated for C_13_H_17_O_4_ [M+H]^+^ 237.1121, found 237.1122.

#### 4.3.5. Compound (**18**)

1-(2,3,4-Trihydroxy-5-(3-methylbut-2-en-1-yl)phenyl)ethan-1-one (**18**). Yield 12.3%, white solid. m.p.: 110–115 °C. ^1^H NMR (300 MHz, CDCl_3_) δ 7.01 (s, 1H), 5.24 (t, *J* = 7.8 Hz, 1H), 3.22 (d, *J* = 7.1 Hz, 2H), 2.48 (s, 3H), 1.70 (s, 3H), 1.67 (s, 3H). ^13^C NMR (75 MHz, CDCl_3_) δ 203.64, 149.74, 149.62, 133.14, 131.12, 122.37, 121.93, 120.01, 112.99, 27.76, 26.11, 25.72, 17.74. HRMS (ESI) *m/z*, calculated for C_13_H_17_O_4_ [M+H]^+^ 237.1121 found 237.1122.

### 4.4. General Procedure for Protection of Hydroxyl Groups (IV)

The 2-Hydroxyacetophenone derivatives (**14**–**18**) (0.6 mmol) were dissolved individually in 3 mL of ACN and K_2_CO_3_ (1.8 mmol) was added, the resulting solutions were stirred for 10 min followed by 2 equivalents of methoxymethyl chloride (ClMOM). The reaction mixtures were brought to 60 °C for 1 h in the microwave. The new compounds **19**–**22** were purified by column chromatography with silica gel 60, eluting Hex:AcOEt (2:1) (*v/v*). The synthesis of **19**–**22** has been achieved in an overall yield of 48.0–70.1%. 

### 4.5. General Procedure for Preparation of Prenylated Chalcones (VI)

The synthesis of chalcones was achieved according to the previously reported procedures for the Claisen−Schmidt reaction (**4**) [12,13]. Briefly, methoxy-methylated acetophenone (**19**) (1 mmol) and benzaldehyde derivatives (1.05 mmol) were dissolved in ethanol (10 mL), the resulting reaction mixtures were kept at room temperature and magnetic stirring for 5 min. Then, a KOH/EtOH solution (1.1 mmol on 10 mL) was added dropwise and stirring was continued at 40 °C for 12 h. The compounds **27**–**28** were purified using column chromatography with silica gel 60, eluting with Hex: AcOEt (2:1) (*v*/*v*). The synthesis of prenylated chalcones has been achieved in an overall yield of 46.3–77.5%.

#### 4.5.1. Compound (**27**)

1-(2-Hydroxy-4-(methoxymethoxy)-5-(3-methylbut-2-en-1-yl)phenyl)-3-(4-methoxyphenyl)prop-2-en-1-one (**27**). Yield 56.3%, white solid. m.p.: 108–109 °C. ^1^H NMR (300 MHz, CDCl_3_) δ 13.38 (s, 1H), 7.91 (d, *J* = 15.4 Hz, 1H), 7.67 (d, *J* = 6.8 Hz, 3H), 7.50 (d, *J* = 15.3 Hz, 1H), 7.01 (d, *J* = 8.3 Hz, 2H), 6.70 (s, 1H), 5.35–5.30 (m, 3H), 3.92 (s, 3H), 3.53 (s, 3H), 3.34 (d, *J* = 7.3 Hz, 2H), 1.80 (s, 6H). ^13^C NMR (75 MHz, CDCl_3_) δ 192.01, 164.81, 161.79, 161.35, 144.15, 132.84, 130.38, 130.07, 126.10, 122.45, 121.92, 118.00, 114.50, 114.29, 102.18, 93.98, 56.39, 55.49, 28.44, 25.84, 17.90. HRMS (ESI) *m/z*, calculated for C_23_H_26_O_5_Na [M+Na]^+^ 405.1672, found 405.1673.

#### 4.5.2. Compound (**28**)

3-(4-Chlorophenyl)-1-(2-hydroxy-4-(methoxymethoxy)-5-(3-methylbut-2-en-1-yl)phenyl)prop-2-en-1-one (**28**). Yield 77.5%, white solid. m.p.: 103–105 °C. ^1^H NMR (300 MHz, CDCl_3_) 13.18 (s, OH), 7.80 (d, *J* = 15.5 Hz, 1H), 7.59 (d, *J* = 5.9 Hz, 2H), 7.53 (d, *J* = 16.8 Hz, 2H), 7.40 (d, *J* = 8.5 Hz, 2H), 6.65 (s, 1H), 5.37–5.19 (m, 3H), 3.48 (s, 3H), 3.29 (d, *J* = 7.1 Hz, 2H), 1.75 (s, 6H). ^13^C NMR (75 MHz, CDCl_3_) δ 191.62, 165.00, 161.69, 142.73, 136.52, 133.38, 132.90, 130.09, 129.68, 129.31, 122.37, 122.18, 120.95, 114.15, 102.20, 93.98, 56.41, 28.46, 25.84, 17.94. HRMS (ESI) *m/z*, calculated for C_22_H_23_ClO_4_Na [M+Na]^+^ 409.1177, found 409.1135.

### 4.6. General Procedure for Preparation of Prenylated Flavones (VII)

The hydroxychalcones (**27**–**28**) (0.5 mmol) were dissolved in DMSO (5 mL) and these solutions were treated with a catalytic amount of iodine [14]. The resulting mixtures were charged in 10 mL glass tubes containing a magnetic stirring bar and a rubber cap. The tubes were subjected to MW at 120 °C for 5 min. After completion of the reaction, the tubes were removed, cooled to room temperature, and the mixture was purified by column chromatography with silica gel 60, eluting Hex:AcOEt (2:1) (*v/v*). The synthesis of prenylated flavones (**29**–**30**) has been achieved in an overall yield of 7.5–41.3%. 

#### 4.6.1. Compound (**29**)

7-(Methoxymethoxy)-2-(4-methoxyphenyl)-6-(3-methylbut-2-en-1-yl)-4H-chromen-4-one (**29**). Yield 41.3%, white solid. m.p.: 70.5 °C. ^1^H NMR (300 MHz, CDCl_3_) δ 7.98 (s, 1H), 7.88 (d, *J* = 8.9 Hz, 2H), 7.20 (s, 1H), 7.02 (d, *J* = 11.8 Hz, 2H), 6.77 (s, 1H), 5.39–5.26 (m, 3H), 3.89 (s, 3H), 3.52 (s, 3H), 3.40 (d, *J* = 7.4 Hz, 2H), 1.74 (m, 6H). ^13^C NMR (75 MHz, CDCl_3_) δ 178.02, 163.4, 162.37, 159.52, 156.33, 133.44, 129.70, 128.01, 125.59, 124.10, 121.44, 117.7, 114.46, 105.77, 101.51, 94.29, 56.40, 55.56, 28.62, 25.92, 17.88. HRMS (ESI) *m/z*, calculated for C_23_H_25_O_5_ [M+H]^+^ 381.1696, found 381.1704.

#### 4.6.2. Compound (**30**)

2-(4-Chlorophenyl)-7-(methoxymethoxy)-6-(3-methylbut-2-en-1-yl)-4H-chromen-4-one (**30**). Yield 30.1%, white solid. m.p.: 108–109 °C. ^1^H NMR (300 MHz, Acetone) δ 8.10 (d, *J* = 8.7 Hz, 2H), 7.86 (s, 1H), 7.62 (d, *J* = 8.7 Hz, 2H), 7.34 (s, 1H), 6.81 (s, 1H), 5.45 (s, 2H), 5.35 (t, *J* = 7.9 Hz, 1H), 3.51 (s, 3H), 3.41 (d, *J* = 7.4 Hz, 2H), 1.75 (s, 6H). ^13^C NMR (75 MHz, Acetone) δ 177.13, 162.07, 160.17, 156.85, 137.58, 133.58, 131.48, 130.15, 129.92, 128.58, 125.66, 122.43, 118.53, 107.90, 102.45, 95.01, 56.43, 28.1, 25.77, 17.71. HRMS (ESI) *m/z*, calculated for C_22_H_22_ClO_4_ [M+H]^+^ 385.1201, found 385.1203.

### 4.7. General Procedure for Preparation of Chromones (II and V) 

The series of compounds **5**–**9** and **19**–**22** (1 mmol) were suspended in triethyl orthoformate (TEOF) (0.5 mL), the resulting solutions were treated with 70% HClO_4_ (0.01 mL, 0.17 mmol) slowly [15]. The mixtures were then stirred at room temperature for 2 h, and ether (200 mL) was added, subsequently, the solution was filtered and the solid was purified by column chromatography with silica gel 60, eluting Hex:AcOEt (2:1) (*v/v*). The synthesis of chromones (**10**–**13**) and prenylated chromones (**23**–**26**) was achieved in an overall yield of 30.4–70.2% and 8.21–40.1% respectively.

#### 4.7.1. Compound (**10**)

7-Hydroxy-4H-chromen-4-one (**10**). Yield 56.9%, white solid, m.p.: 215 °C. ^1^H NMR (300 MHz, Acetone) δ 8.03 (d, *J* = 6.0 Hz, 1H), 7.97 (d, *J* = 8.7 Hz, 1H), 6.98 (d, *J* = 9.8 Hz, 1H), 6.89 (s, 1H), 6.19 (d, *J* = 6.0 Hz, 1H). ^13^C NMR (75 MHz, Acetone) δ 175.81, 162.51, 158.25, 155.53, 127.00, 117.95, 114.75, 112.21, 102.48. HRMS (ESI) *m/z*, calculated for C_9_H_7_O_3_ [M+H]^+^ 163.0389, found 163.0388.

#### 4.7.2. Compound (**11**)

6-Hydroxy-4H-chromen-4-one (**11**). Yield 30.55%, white solid, m.p.: 242–243 °C. ^1^H NMR (300 MHz, DMSO) δ 10.07 (s, OH), 8.2 (d, *J* = 6.0 Hz, 1H), 7.50 (d, *J* = 9.0 Hz, 1H), 7.29 (d, *J* = 3.0 Hz, 1H), 7.22 (dd, *J* = 9.0, 3.0 Hz, 1H), 6.25 (d, *J* = 6.0 Hz, 1H). ^13^C NMR (75 MHz, DMSO) δ 176.47, 156.68, 154.89, 149.84, 125.23, 123.25, 119.97, 111.21, 107.58. HRMS (ESI) *m/z*, calculated for C_9_H_7_O_3_ [M+H]^+^ 163.0389, found 163.0399. 

#### 4.7.3. Compound (**12**)

5-Hydroxy-4H-chromen-4-one (**12**). Yield 40.1%, white solid, m.p.: 125–126 °C. ^1^H NMR (300 MHz, Acetone) δ 12.57 (s, 1H), 8.21 (d, *J* = 5.9 Hz, 1H), 7.64 (t, *J* = 8.4 Hz, 1H), 6.99 (d, *J* = 8.4 Hz, 1H), 6.78 (d, *J* = 8.2 Hz, 1H), 6.35 (d, *J* = 5.9 Hz, 1H). ^13^C NMR (75 MHz, Acetone) δ 183.03, 160.90, 157.80, 156.81, 135.72, 111.51, 111.14, 110.98, 107.16. HRMS (ESI) *m/z*, calculated for C_9_H_7_O_3_ [M+H]^+^ 163.0389, found 163.0389.

#### 4.7.4. Compound (**13**)

7,8-Dihydroxy-4H-chromen-4-one (**13**). Yield 34.1%, white solid, m.p.: 265 °C. ^1^H NMR (300 MHz, MeOD) δ 8.11 (d, *J* = 5.9 Hz, 1H), 7.52 (d, *J* = 8.8 Hz, 1H), 6.95 (d, *J* = 8.8 Hz, 1H), 6.26 (d, *J* = 5.9 Hz, 1H). ^13^C NMR (75 MHz, MeOD) δ 178.61, 156.12, 150.59, 147.22, 132.94, 117.50, 115.44, 113.98, 110.88. HRMS (ESI) *m/z*, calculated for C_9_H_7_O_4_ [M+H]^+^ 179.0338 found 179.0338.

#### 4.7.5. Compound (**23**)

7-Hydroxy-6-(3-methylbut-2-en-1-yl)-4H-chromen-4-one (**23**). Yield 17.32%, white solid. m.p.: 120–125 °C. ^1^H NMR (300 MHz, CDCl_3_) δ 8.73 (s, 1H), 7.96 (s, 1H), 7.79 (d, *J* = 6.0 Hz, 1H), 6.97 (s, 1H), 6.32 (d, *J* = 6.0 Hz, 1H), 5.35 (t, *J* = 7.3 Hz, 1H), 3.43 (d, *J* = 7.2 Hz, 2H), 1.74 (s, 6H). ^13^C NMR (75 MHz, CDCl_3_) δ 178.22, 160.98, 157.13, 155.31, 134.90, 128.19, 126.08, 120.98, 117.66, 112.18, 102.82, 28.85, 25.86, 17.91. HRMS (ESI) *m/z*, calculated for C_14_H_15_O_3_ [M+H]^+^ 231.1015, found 231.1023.

#### 4.7.6. Compound (**24**)

7-Hydroxy-8-(3-methylbut-2-en-1-yl)-4H-chromen-4-one (**24**). Yield 21.14%, white solid. m.p.: 172–173 °C. ^1^H NMR (300 MHz, CDCl_3_) δ 8.44 (s, 1H), 7.96 (d, *J* = 8.8 Hz, 1H), 7.88 (d, *J* = 5.9 Hz, 1H), 7.04 (d, *J* = 8.8 Hz, 1H), 6.32 (d, *J* = 5.9 Hz, 1H), 5.26 (t, *J* = 7.2 Hz, 1H), 3.58 (d, *J* = 7.1 Hz, 2H), 1.83 (s, 3H), 1.71 (s, 3H). ^13^C NMR (75 MHz, CDCl_3_) δ 178.51, 160.14, 156.21, 155.43, 134.13, 124.55, 120.92, 118.10, 115.37, 115.14, 112.02, 25.82, 22.28, 17.98. HRMS (ESI) *m/z*, calculated for C_14_H_15_O_3_ [M+H]^+^ 231.1015, found 231.1012.

#### 4.7.7. Compound (**25**)

5-Hydroxy-6-(3-methylbut-2-en-1-yl)-4H-chromen-4-one (**25**). Yield 5.21%, white solid. m.p.: 235–236 °C. ^1^H NMR (300 MHz, CDCl_3_) δ 12.67 (s, 1H), 7.81 (d, *J* = 5.9 Hz, 1H), 7.42 (d, *J* = 8.7 Hz, 1H), 6.85 (d, *J* = 8.6 Hz, 1H), 6.26 (d, *J* = 5.9 Hz, 2H), 5.30 (dd, *J* = 14.6, 7.1 Hz, 1H), 3.36 (d, *J* = 7.2 Hz, 2H), 1.74 (d, *J* = 7.5 Hz, 6H). ^13^C RMN (75 MHz, CDCl_3_) δ 183.35, 158.90, 157.89, 156.22, 135.55, 133.55, 124.20, 121.56, 111.35, 111.10, 106.48, 27.06, 25.83, 17.85. HRMS (ESI) *m/z*, calculated for C_14_H_15_O_3_ [M+H]^+^ 231.1015, found 231.1001.

#### 4.7.8. Compound (**26**)

7,8-Dihydroxy-6-(3-methylbut-2-en-1-yl)-4H-chromen-4-one (**26**). Yield 8.21%, white solid. m.p.: 255–258 °C. ^1^H NMR (300 MHz, Acetone) δ 7.86 (d, *J* = 6.0 Hz, 1H), 7,27 (s, 1H), 5.98 (d, *J* = 6.0 Hz, 1H), 5.26 (t, *J* = 7.4 Hz, 1H), 3.29 (d, *J* = 7.7 Hz, 2H), 1.60 (s, 6H). ^13^C NMR (75 MHz, Acetone) δ 175.96, 154.60, 148.54, 141.61, 132.59, 132.39, 126.80, 122.07, 117.58, 114.64, 111.78, 26.72, 25.04, 16.94. HRMS (ESI) *m/z*, calculated for C_14_H_15_O_4_ [M+H]^+^ 247.0964, found 247.0921.

### 4.8. Cytotoxic Activity

Cytotoxicity was evaluated in the human monocyte cell line U-937 (ATCC CRL-1593.2) at the exponential growth phase, adjusted at 1 × 10^5^ cells/mL in complete RPMI-1640 medium (RPMI-1640 enriched with 200 mM L-glutamine, 10% inactivated fetal bovine serum (FBS) and 1% of a mixture of 10,000 IU/mL penicillin plus 10,000 mg/mL streptomycin). Then, in each well of a 96-well tissue culture plate were dispensed 100 mL of cells plus 100 mL of each compound (as one of six serials 1:2 dilution concentrations starting at 368 µM prepared in the same medium). Doxorubicin was included as an internal positive control under the same dilution pattern, starting at 18 µM whereas unexposed cells were used as negative controls. Afterward, cells were incubated for 72 h at 37 °C and 5% CO_2_. Cell viability was assayed by the MTT reduction assay according to the optical density (O.D) at 570 nm of the resulting reduction of formazan in a Varioskan Flash Multimode Reader (Thermo Scientific, Waltham, MA, USA) [18]. Nonspecific absorbance was corrected by subtracting the O.D of the blank solution that corresponded to complete RPMI-1640 medium. The assay was done in triplicate in at least two independent experiments. 

### 4.9. Anti-Trypanosomal Activity

This activity was carried out in intracellular amastigotes of *T. cruzi* (Tulahuen strain transfected with the β-galactosidase gene [19]. Briefly, metacyclic trypomastigotes were cultured at 26 °C for ten days in a modified NNN (Novy−McNeal−Nicolle) medium. U937 cells were seeded in 96-well tissue culture plates at a density of 2.5 × 10^4^ cells in 100 μL of complete RPMI-1640 medium/well and exposed for 24 h to phorbol myristate acetate (1 ng/mL) to induce transformation of monocytes into macrophages. Cells were infected with trypomastigotes (5 parasites per cell) and incubated for 24 h at 37 °C, 5% CO_2_. Wells were washed twice with warm phosphate buffer solution (PBS) to remove noninternalized parasites, and then were added 100 μL of complete RPMI-1640 medium and 100 μL of each concentration of compound (50, 12.5, 3.125 μM). BNZ, at the same concentrations, was used as internal control for trypanocidal activity (positive control), and nontreated cells as controls for infection (negative control). After 72 h of incubation 37 °C, 5% CO_2_, the viability of intracellular amastigotes was determined by measuring the β-galactosidase activity. For this, 100 μM of Chlorophenol red-β-D- galactopyranoside (CPRG), and 0.1% Nonidet P-40 was added to each well and incubated for 4 h at 37 °C, at 24 °C protected from light. After that, measurement β-galactosidase activity was measured at 570 nm on a Varioskan, Thermo spectrophotometer. Nonspecific absorbance was subtracted from the measurement. Infected cells exposed to benznidazole were used as controls for anti-trypanosomal activity. Determinations were done in triplicate with at least two independent experiments [18].

### 4.10. Data Analysis

The cytotoxicity was expressed as the medial lethal concentration (LC_50_), while the trypanocidal activity was defined as the median effective concentration (EC_50_) according to the percentage of inhibition of cells and parasites, respectively, as described elsewhere [19]. Both LC_50_ and EC_50_ were calculated by the Probit analysis. The cytotoxicity level was graded based on the own hit criteria into high when LC_50_ values were lower than 50 μM, moderate when LC_50_ was higher than 50 μM and lower than 100 μM, and, low when LC_50_ was higher than 100 μM. Similarly, the trypanocidal activity was graded according to EC_50_ values, based on the hit criteria proposed by others [20] into high when EC_50_ was lower than 10 μM, moderate when EC_50_ values were higher than 10 μM but lower than 50 μM and, low when EC_50_ was higher than 50 μM. The relation between cytotoxicity and trypanocidal activity was expressed as the index of selectivity (IS) that corresponded to the ratio when dividing the LC_50_ by the EC_50_.

## Data Availability

The data presented in this study are available in this article or Appendix A.

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
