# Peer review of "Synthesis and Evaluation of Trypanocidal Activity of Chromane-Type Compounds and Acetophenones"

_molecules, 2021, doi:10.3390/molecules26237067_

Round 1
Reviewer 1 Report
In this manuscript the authors reported the synthesis and the anti-trypanosoma activity of acetophenone and chalcone derivatives and chromane-type compounds.
Table 1 can be improved reporting the chemical structures of compounds to make the results clearer to the readers
Author Response
Evaluator 1
R/. The general structures of the synthesized compounds were included in Table 1 for easy reading.
Please see the attachment

Reviewer 2 Report
Luis A. González et al. describe the synthesis of pyran-4-ones substituted with hydroxyl and isoprenyl residues and their activity against Trypanosoma cruzi.
Overall, I am quite wary of the biological activities of some of these compounds. For example, compound 8, which the authors correctly identify as the most promising one based on its Selectivity Index, is a polyphenol, which are prominent members of the Pan-Assay Interference Compounds (PAINS).
Regarding the structural characterization of the compounds, the authors do not provide an assigment of the 13C data. Also, there is no 13C and bidimensional spectra in the suplementary information, only 1H. I consider that an assigment of the 13C data in the article, and 13C and bidimensional spectra in the suplementary information are a must for publication.
Looking into the spectroscopic data provided, there are some points that would require some explanation/correction by the authos:
- Compund 5. The 1H NMR spectra in the SI shows hydrogens 3´and 5´ together integrating for only 1.21.
- Compund 7. The 1H NMR spectra in the SI shows hydrogens 3´and 5´ together integrating for only 1.05.
- Compound 18. The hydrogen at 7 ppm is not 5', should be 6' (figure in the SI).
- Compound 25. A total of 13 carbons are listed in the 13C data (lines 356, 357 of the manuscript), but 14 carbons should be expected. It is likely that some overlapping is obscuring one of the carbons. This is one of the reasons why 13C assigments should be required.
- Compound 26. The integration of the hydrogen 5 is missing in the spectra (SI).
- Compound 27. The numbering of the hydrogens of the spectra provided in the SI seems to be incorrect.
Overall, taking into account my reticence with the biological activity presented by polyphenols, the lack of 13C data assigment, and the issues found with some of the spectroscopic data presented (points 1 to 6 above), I can not endorse the publication of the manuscript in Molecules.
